# Can we Defend Against the Unknown? An Empirical Study About Threshold Selection for Neural Network Monitoring

**Khoi Tran Dang**[1, 2]        **Kevin Delmas**[3]        **Jérémie Guiochet**[2]        **Joris Guérin**[2, 4]

[1]INSA Toulouse    [2]LAAS-CNRS, Univ. Toulouse    [3]ONERA, Toulouse, France
[4]Espace-Dev, IRD, Univ. Montpellier, Montpellier, France
*tkdang@insa-toulouse.fr ; kevin.delmas@onera.fr ; jeremie.guiochet@laas.fr ; joris.guerin@ird.fr*

## Abstract

With the increasing use of neural networks in critical systems, runtime monitoring becomes essential to reject unsafe predictions during inference. Various techniques have emerged to establish rejection scores that maximize the separability between the distributions of safe and unsafe predictions. The efficacy of these approaches is mostly evaluated using threshold-agnostic metrics, such as the area under the receiver operating characteristic curve. However, in real-world applications, an effective monitor also requires identifying a good threshold to transform these scores into meaningful binary decisions. Despite the pivotal importance of threshold optimization, this problem has received little attention. A few studies touch upon this question, but they typically assume that the runtime data distribution mirrors the training distribution, which is a strong assumption as monitors are supposed to safeguard a system against potentially unforeseen threats. In this work, we present rigorous experiments on various image datasets to investigate: 1. The effectiveness of monitors in handling unforeseen threats, which are not available during threshold adjustments. 2. Whether integrating generic threats into the threshold optimization scheme can enhance the robustness of monitors.

## 1 INTRODUCTION

Deep learning has gained traction in safety-critical domains such as surgical robots [Haidegger, 2019], autonomous vehicles [Ferreira et al., 2022], and drone landing [Guerin et al., 2022a]. As reliance on neural networks (NN) in these sectors intensifies, the importance of ensuring their safety keeps growing and demands continued research. NN runtime monitoring is a promising direction, seeking to detect unsafe predictions during inference. Numerous methods have been developed for NN runtime monitoring [Hendrycks and Gimpel, 2016, Ferreira et al., 2023, Wang et al., 2022]. They consist of designing scoring functions indicating the level of confidence for a prediction. These scores are then thresholded to reject low-confidence predictions.

The performance of a monitor is assessed based on its capacity to build score distributions that effectively separate safe and unsafe predictions. To evaluate this, commonly used metrics in the literature are threshold-agnostic, representing an average performance of binary classification metrics across a range of threshold values (e.g., area under the receiver operating characteristic curve (AUROC)). High values of such metrics suggest the existence of a good threshold, but they do not ensure that it can be found easily. To deploy a monitor in a real-world application, a concrete rejection threshold value must be set to determine accepted and rejected predictions. This threshold is pivotal, as a good monitor with a poor threshold can still result in an unsafe system. Despite the crucial nature of threshold optimization, it remains under-explored in runtime monitoring research.

Building upon the foundational work of Chow [1970], the field of "classification with rejection" has considered the problem of rejection thresholds optimization [Geifman and El-Yaniv, 2017, Zhang et al., 2023]. However, a strong assumption underlying most of these studies is that the data distribution encountered during runtime closely mirrors the training distribution. In practice, this means that the validation occurs on training and test datasets drawn from the same distribution. This presents a notable challenge in the context of neural network safety monitoring, where the primary objective is to safeguard critical systems against various types of threats, such as novel classes, covariate shifts, or adversarial attacks. In this paper, we aim to assess experimentally the resilience of runtime monitoring thresholds under different assumptions about our prior knowledge of runtime threats. This includes investigating scenarios that depart from the traditional assumption of distributional similarity, offering a broader coverage of diverse real-world conditions.

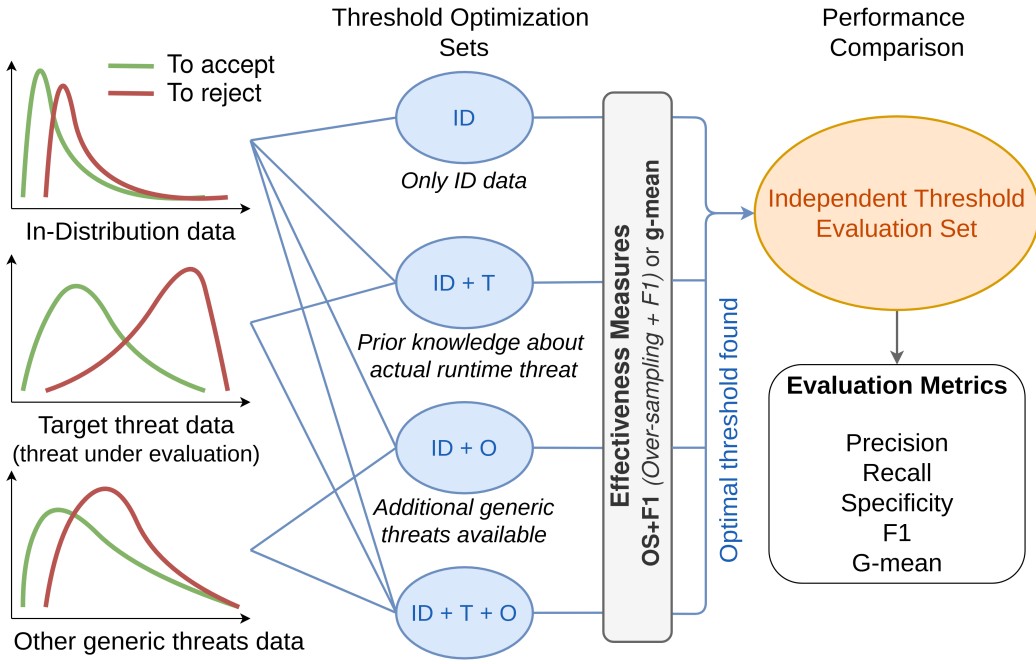

Figure 1: **Conceptual Overview** – This research compares four ways to construct threshold optimization sets for neural network runtime monitors, each representing distinct assumptions about the data available for threshold tuning.

To address this pivotal question, we have designed a rigorous large-scale experiment on computer vision datasets. We compare four different ways to construct a threshold optimization dataset (see Figure 1 and Section 3), allowing us to investigate two primary research questions. First, we compare thresholds fitted with or without prior knowledge of the evaluated threat, which is a more realistic setting to determine whether NN monitors can effectively handle unforeseen new threats. Second, we explore the potential benefits of integrating generic threats into the threshold optimization dataset. Given the relative ease of generating or acquiring generic threats, this approach could represent a realistic straightforward method to enhance the robustness of neural network monitors.

This paper is organized as follows: Section 2 reviews relevant literature on NN monitoring and threshold optimization for classification. Section 3 presents our methodology and the associated research questions. Section 4 outlines our experimental design. Section 5 analyzes our findings. In Section 6, we reflect on our findings and explore their practical implications. Finally, in Section 7, we conclude this work and suggest future research directions.

## 2 BACKGROUND AND RELATED WORK

In this section, we present key definitions and relevant literature about NN runtime monitoring. This work focuses on classification, but some of the methodologies discussed here are transferable to other Machine Learning tasks.

### 2.1 NEURAL NETWORKS RUNTIME MONITORS

Let us denote a classification task by $T$, its feature space by $\mathcal{X}$, and its label space by $\mathcal{Y}$. The oracle function for $T$ is denoted $\Omega$, signifying that the ground truth for any $x \in \mathcal{X}$ is $\Omega(x)$. Let $D_{\text{train}}$ represent a training dataset for $T$, and let $f$ be a classifier for $T$, trained using $D_{\text{train}}$. A runtime monitor for $f$, denoted as $m_f$, is a binary classifier designed to filter out unsafe predictions of $f$. Here, we adopt a convention where the positive class for $m_f$ denotes unsafe samples, though the reverse convention also exists in the literature.

Most of the literature on NN monitoring does not focus on constructing binary classifiers, but rather on models that output continuous scores representing the confidence in a prediction. In practice, training a monitor, i.e., adjusting the parameters of the monitoring function to generate meaningful scores, commonly involves the use of the same labeled training dataset, $D_{\text{train}}$, although this is not a strict requirement. Converting these scores into binary classification outputs requires applying a thresholding operation.

The fitting method typically relies on features extracted from one or more layers of $f$. Hendrycks and Gimpel [2016] proposed to detect abnormal examples using the maximum softmax probability (MSP) as their score. Lee et al. [2018] fitted class-conditional Gaussian distributions to the features and defined their confidence score as the minimum Mahalanobis distance to class-wise centroids. Henzinger et al. [2020] compared runtime features to the smallest bounding boxes containing features from $D_{\text{train}}$. Liu et al. [2020] pro-

posed the energy score (*logsumexp* of the logits) and Sun et al. [2021] suggested computing rectified logits by clipping the activations. Recently, Wang et al. [2022] developed a virtual logits score, generated from the norm of feature residuals against the principal subspace defined by $D_{\text{train}}$.

## 2.2 EVALUATION OF NN RUNTIME MONITORS

### 2.2.1 Out-of-Distribution vs. Out-of-Model-Scope

The concept of *safety* is central in defining runtime monitors expected outcomes. Two perspectives coexist to define what constitutes an unsafe sample [Guerin et al., 2023]:

1. Out-of-Distribution (OOD): This perspective targets the detection of data points that fall beyond the training distribution of the classifier, represented by $D_{\text{train}}$.

2. Out-of-Model-Scope (OMS): This perspective focuses on identifying data points that lead to incorrect predictions by the classifier.

In this study, we adopt the OMS approach, where the monitor's objective is to reject misclassified samples, indicated as $m_f = 0$ when $f(x) = \Omega(x)$ (correct prediction) and $m_f = 1$ when $f(x) \neq \Omega(x)$ (misclassification). As explained by Guerin et al. [2023], the OMS setting circumvents the potentially ambiguous definition of what is OOD and avoids any misconceptions about OOD detection performance. It's important to note that the training dataset $D_{\text{train}}$, traditionally considered in-distribution in the OOD setting, often contains OMS (misclassified) samples since classifiers are rarely perfect. In summary, our study defines a good monitor as one that rejects incorrect predictions and accepts correct ones, regardless of whether the corresponding samples are considered in or out-of-distribution.

### 2.2.2 Evaluation Dataset Construction

Even in the OMS setting, it's crucial to evaluate a monitor's performance outside the training distribution, where misclassifications are more likely. Hence, in typical evaluations of monitors, in-distribution (ID) data and out-of-distribution (OOD) threat data are used jointly to assess performance. For ID test data, we usually use the test split associated with $D_{\text{train}}$. Threat data primarily fall into three categories: 1. *Novelty*: The labels do not belong to the label space ($\Omega(x) \notin \mathcal{Y}$), 2. *Covariate Shift*: The inputs are not drawn from the same distribution as $D_{\text{train}}$, 3. *Adversarial Attacks*: The inputs are maliciously modified to cause misclassifications. In the OMS setting, we use labeled datasets to identify errors of $f$ to serve as ground truth for the monitor evaluation. Both the test and threat sets may contain misclassifications. Additionally, except for novelty, the threat sets can contain correct predictions, depending on the degree of perturbations.

### 2.2.3 Threshold Agnostic Evaluation Metrics

A monitor is evaluated based on its ability to distinguish correctly classified data from misclassifications. Related works frequently use threshold-agnostic metrics to assess this skill across a range of thresholds. Examples of such metrics include AUROC, AUPR (Area under the Precision-Recall curve), and FNR@95TNR (False Negative Rate at 95% True Negative Rate). However, to deploy a runtime monitor in a real-world scenario, one must select a fixed threshold value to decide which predictions to reject. As of today, no studies have addressed the generic problem of threshold selection for neural network monitoring. Threshold selection is typically addressed in a somewhat nebulous manner, suggesting that the "threshold should be chosen such that a high proportion of ID data instances are accurately processed by the monitor" [Liu et al., 2020, Sun et al., 2021, Wang et al., 2022].

## 2.3 THRESHOLD OPTIMIZATION FOR CLASSIFICATION

Despite the absence of work addressing threshold fitting for NN runtime monitoring, some research has tackled this problem in the broader context of classification. Arampatzis and van Hameran [2001] explained the steps involved in the exhaustive search method for threshold optimization on a finite test dataset: 1. Calculate the classification scores for all samples of the test dataset, 2. Sort the list of predicted scores, 3. Select a metric to represent threshold performance, called *effectiveness measure*, 4. Calculate the effectiveness measure at every position of the sorted list, 5. Find the position where the effectiveness measure is optimal, 6. Set the threshold slightly above this optimal position.

In the literature, the most common variations of this standard optimization pipeline involve alternative choices for the effectiveness measure: F-score [Zou et al., 2016], geometric mean of Recall and Specificity [Johnson and Khoshgoftaar, 2021], Matthews correlation coefficient [Chicco and Jurman, 2023], or Cohen's kappa [Freeman and Moisen, 2008]. Another research direction involves developing optimized search strategies to identify the threshold more efficiently [Arampatzis and van Hameran, 2001, Esposito et al., 2021].

In this study, we compare four ways to construct the validation set used to optimize the threshold for runtime monitors.

## 3 METHODOLOGY

Let us consider a monitor, that has been trained to produce scores reflecting the confidence of a NN. Our goal is to compare different ways to build a validation dataset on which we can find an optimal threshold for these scores, to determine the predictions to reject. Although the process of

finding a suitable threshold has received little attention in the literature, it is a crucial factor to consider. In practice, a monitor may generate scores that accurately distinguish incorrect predictions, but its safety could be compromised if the rejection threshold is not properly calibrated.

To evaluate the effectiveness of a given threshold, we employ conventional binary classification metrics, such as Recall and Precision, on a carefully designed test dataset, which we call *Threshold Evaluation Set*. To have a balanced evaluation, we construct the Threshold Evaluation Set to encompass regular in-distribution data as well as one specific target threat. The inclusion of in-distribution data enables us to identify monitors that may overly reject, and focusing on a single threat allows us to characterize distinct monitor failures. This focus is more realistic, as it is unlikely for a NN to encounter multiple threats concurrently. We emphasize that our experiments address multiple threats in practice, but they are assessed separately to evaluate monitor performance across different threat scenarios.

To tune the threshold, we use a separate *Threshold Optimization Set*. Fitting the threshold essentially involves identifying the value that optimizes a specific effectiveness measure on the Optimization Set (see Section 2.3). The chosen effectiveness measure should reflect the delicate balance between system safety and availability, i.e., it should encapsulate the monitor's capacity to reject incorrect predictions and to accept correct ones [Guerin et al., 2022b]. In our experiments, we try F1 and g-mean (see Section 4). Both the Threshold Optimization and Evaluation sets are composed of inputs to the NN (images), corresponding monitor scores, and labels that indicate the correctness of the predictions.

Our experiments compare four ways to construct the Optimization Set (Figure 1). They reflect alternative real-world deployment scenarios for monitors, representing assumptions about our ability to anticipate forthcoming threats:

1. The first assumption, denoted ID, involves constructing an optimization set composed exclusively of In-Distribution (ID) data samples. This presumes that no threat data is accessible for threshold optimization. In the remaining approaches, ID samples are still present, along with other samples corresponding to threats.

2. The second approach, denoted ID+T, involves enriching the optimization set with data samples associated with the Target threat (T), i.e., the threat under evaluation. This scenario corresponds to situations where threats pertinent to the system have been previously identified, such as through a system safety analysis.

3. The third approach, denoted ID+O, designs an optimization set without the target threat, but including samples corresponding to Other generic threats (O). This scenario examines if awareness of generic threats can aid in determining a more effective threshold for unanticipated, new threats.

4. The fourth approach, denoted ID+T+O, employs an optimization set containing data samples for both the Target and Other generic threats. It aims to assess the performance of a monitoring threshold when multiple threats are used and one of them is the target threat.

A summary of how the Optimization and Evaluation sets are constructed for the different approaches can be found in Table 1. It shows that the Evaluation Set is always the same and never overlaps with the Optimization set.

The objective of comparing these four approaches is twofold. First, we aim to assess the effectiveness of monitors when the target threat T is unknown, which reflects a more realistic scenario. This evaluation helps us understand if monitors, as evaluated in previous literature using threshold-agnostic metrics or optimization sets mirroring the training distribution, can be relied upon in real-world situations to protect systems from unknown threats. Our comparison aims to determine whether experiments from previous works are sufficient to draw conclusions about a monitor's real-world performance or if additional tests are needed before deployment. As a result, we formulate our first research question as: *RQ1 – Can we obtain similar monitoring performance without assuming prior knowledge of runtime threats during threshold tuning?*

To answer RQ1, we compare ID against ID+T, and ID+O against ID+T+O. If our findings reveal that prior awareness about the evaluated threat is crucial, it could significantly limit the applicability of runtime monitors. Indeed, the main objective of monitoring is to address unforeseen hazards. If knowledge about the actual threats that an NN will encounter is readily available, such examples would typically be incorporated during training. It is worth noting that several studies have used this strategy for tuning monitor hyperparameters by simply dividing the evaluation set into validation and test subsets [Hsu et al., 2020].

The second objective is to evaluate whether the strategy of adding a pool of generic threat data to tune the threshold can be viable to increase the robustness of the monitor to unforeseen threats. Such generic threats are easy to obtain by collecting additional image data from the internet or adding perturbations to ID data. On the one hand, adding such generic threat data can help generalization by adding difficult examples to better delineate the boundaries of what the NN knows. On the other, it could also be detrimental if the selected generic threats are too diverse. Hence, our second objective is to answer the following research question: *RQ2 – How helpful is the inclusion of generic threats data?*

For RQ2, we compare strategy ID against ID+O, as well as ID+T against ID+T+O. A positive answer would be promising, given the relative ease of constructing a generic dataset of threats, which could be utilized to enhance monitoring system performance and subsequently facilitate the adoption of neural networks in safety-critical systems.

Table 1: **Threshold Optimization and Evaluation sets** – Methodology to construct the threshold optimization and evaluation sets for the different strategies considered in this study. Set 1 and Set 2 always denote non-overlapping splits of a dataset.

| | | In-Distribution | | Target Threat | | Other Generic |
| | | Set 1 | Set 2 | Set 1 | Set 2 | Threats |
|---|---|---|---|---|---|---|
| Threshold Optimization | ID | ✓ | | | | |
| | ID+T | ✓ | | ✓ | | |
| | ID+O | ✓ | | | | ✓ |
| | ID+T+O | ✓ | | ✓ | | ✓ |
| Threshold Evaluation | | | ✓ | | ✓ | |

# 4 EXPERIMENTAL DESIGN

## 4.1 DATASETS, MODELS AND MONITORS

To answer the aforementioned research questions, we conducted extensive experiments. To encapsulate varying ID scenarios, we use three image classification datasets: CIFAR10, CIFAR100 [Krizhevsky et al., 2009] and SVHN [Netzer et al., 2011]. For each ID dataset, we use 2 distinct neural network architectures – DenseNet and ResNet – with weights taken from Lee et al. [2018]. For Densenet, the test accuracies are: CIFAR10 (0.93), CIFAR100 (0.73), SVHN (0.88), and for ResNet: CIFAR10 (0.92), CIFAR100 (0.73), SVHN (0.89).

For each ID dataset and architecture pair, we implement four distinct monitoring techniques. Mahalanobis (Maha) [Lee et al., 2018] and Outside-the-Box (OtB) [Henzinger et al., 2020] are feature-based approaches. We derive the feature representation from the final layer preceding classification and do not apply input pre-processing. On the other hand, Max Softmax Probability (MSP) [Hendrycks and Gimpel, 2016] and Energy (Ene) [Liu et al., 2020] are logit-based methods. Regarding hyperparameters, we use num_box=3 for OtB and T=1 for Ene. These settings resulted in a total of 24 monitors evaluated (3 ID datasets x 2 NNs x 4 monitors).

Each ID set is paired with nine unique threat sets to assess the monitors under varied circumstances:

- 3 novelty sets (datasets with classes distinct from the ID set). For CIFAR 10, the corresponding novelty sets are CIFAR100, SVHN, and LSUN [Yu et al., 2015]. CIFAR100 incorporates CIFAR10, SVHN, and LSUN while SVHN involves CIFAR10, LSUN, and TinyImageNet (a subset of ImageNet [Deng et al., 2009]).

- 3 covariate shifts (transformations from AugLy [Papakipos and Bitton, 2022]), including Brightness (factor=3), Pixelization (ratio=0.5) and Blur (radius=2).

- 3 adversarial attacks (generated with Torchattacks [Kim, 2020]) - FGSM, PGD, and DeepFool using the default settings.

## 4.2 THRESHOLD OPTIMIZATION METHODOLOGY

For each ID dataset–monitor pair, we cycle through the 9 threats, with each serving once as the Target threat (T), resulting in 9 unique outcomes for each optimization set construction approach. While assessing a target threat T, the remaining 8 threats serve as Other Generic Threats (O). The test split of the classifier's training dataset serves as the In-Distribution (ID) dataset. Then, both the ID set and the T set are randomly split in half, so that the threshold evaluation set and the four threshold optimization sets can be constructed, following the methodology presented in Section 3 (Table 1).

To optimize the threshold on the optimization set, we follow the methodology described in Section 2.3. For the effectiveness measure, we initially used F1, the harmonic mean between Precision and Recall, as it is a prevalent choice in the literature. Yet, early experiments revealed that F1 frequently resulted in the unfavorable action of setting exceedingly low thresholds, thereby rejecting all samples in the evaluation set. Of the 864 experiments conducted (24 monitors × 9 threats × 4 optimization sets), this outcome happened 116 times. Such behavior can be attributed to the significant class imbalance often observed in our optimization sets. Indeed, since classifiers typically commit fewer errors with ID data, the ID strategy predominantly contains negative examples (designated for acceptance), and other strategies, notably ID+O and ID+T+O, contain much more positive examples (designated for rejection).

We tested two distinct solutions to address this challenge:

1. over-sampling (OS) the minority class in the threshold optimization set to achieve a positive-to-negative ratio between 0.4 and 0.6,

2. using another effectiveness measure: g-mean, the geometric mean between Recall and Specificity. As Specificity solely considers samples with negative ground truth, g-mean is unaffected by class imbalance. A very low threshold results in a recall of 1 and a specificity of 0, and will not be favored by g-mean optimization.

Both OS+F1 and g-mean approaches are compared in our experiments.

### 4.3 EVALUATION METRICS AND STATISTICAL SYNTHESIS

Once a threshold is chosen, we evaluate its performance on the threshold evaluation set. For each experiment, we compute five evaluation metrics (F1, g-mean, Recall, Precision, Specificity) representing different aspects of the monitor's performance. Computing these diverse metrics allows us to analyze the impact of different threshold optimization approaches more finely.

Given the comprehensive scope of our experiments, we are left with 1728 recorded outcomes for each of these five metrics. Drawing definitive conclusions from such an expansive set of raw results is challenging. Even when we fix the effectiveness measure, we are still tasked with comparing the four threshold optimization approaches across 216 cases. Consequently, we resort to statistical testing to discern the distinctions between approaches across multiple results. Adhering to the methodology outlined by Demšar [2006], we employ the non-parametric Wilcoxon signed-rank tests for comparing two strategies over multiple scenarios ("no difference" null hypothesis, p-value<0.05 for significance). The Friedman test and its associated Nemenyi post-hoc test are utilized for comparing multiple strategies across multiple scenarios.

## 5 RESULTS

The data from our 1728 experiments is complex and not immediately interpretable in its raw form. In this section, we present the outcomes of our statistical analysis and draw associated conclusions. For transparency and reproducibility, the raw results, as well as the code to replicate our experiments have been made available.[1]

### 5.1 COMPARING EFFECTIVENESS MEASURES

First, we compare the two proposed effectiveness measures for threshold tuning on the Optimization set: over-sampling with F1 (OS+F1) and g-mean. For each of the 4 strategies and each of the 5 evaluation metrics, we compare these effectiveness measures using the Wilcoxon signed-rank tests across the 216 experiments. The Wilcoxon test is a non-parametric statistical test, used to compare the performance of two classifiers over multiple datasets [Demšar, 2006]. The results obtained are shown in Table 2. We find that OS+F1 generally yields better Recall and F1 scores, whereas g-mean optimization produces better Precision, Specificity, and g-mean scores. These findings indicate that the choice of the effectiveness measure should be based on the particular metric one seeks to optimize, and this choice should

---

[1] https://github.com/jorisguerin/neural-network-monitoring-benchmark

Table 2: **Effectiveness measures comparison (OS+F1 vs. g-mean)** – Metrics were computed across the 216 experiments, followed by statistical comparison using the Wilcoxon test. The displayed numbers represent p-values, underlined orange text indicates OS+F1 is worse than g-mean, regular blue text indicates OS+F1 is better than g-mean, and italicized black text indicates no significant difference.

|             | ID    | ID+T  | ID+O  | ID+T+O |
|-------------|-------|-------|-------|--------|
| F1          | 3e-08 | 3e-04 | 2e-04 | 8e-06  |
| G-mean      | 1e-26 | 4e-29 | 2e-02 | *5e-02* |
| Recall      | 4e-31 | 2e-32 | 4e-37 | 4e-37  |
| Precision   | 1e-31 | 1e-32 | 9e-36 | 2e-35  |
| Specificity | 1e-31 | 2e-32 | 3e-37 | 3e-37  |

be aligned with the objectives of the system under test. A higher Recall corresponds to a more conservative system, i.e., fewer false acceptances from the monitor. Conversely, higher Precision and Specificity indicate an improved system availability, i.e., fewer false rejections from the monitor. More results comparing effectiveness measures can be found in Appendix A.

### 5.2 COMPARING THRESHOLD OPTIMIZATION SET CONSTRUCTION APPROACHES

Next, we compare the monitoring performance obtained with the different approaches to construct the Threshold Optimization set. To compare several approaches across experiments, we use the Friedman test and its corresponding Nemenyi post-hoc test, as recommended by Demšar [2006]. The Friedman test is a non-parametric test comparing the average ranks of different models, with the null hypothesis assuming no significant difference between them. If the null hypothesis is refuted, the Nemenyi post-hoc test is then applied to identify which model has greater performance.

More precisely, we compare the values obtained for the F1 and g-mean scores on the Threshold Evaluation sets. We focus on these metrics because they are both intended to represent a balance between over-rejection and over-acceptance. At the significance level of $\alpha = 0.05$, the Friedman test shows a significant difference in performance between the four threshold optimization approaches. The results of the Nemenyi test, with both OS+F1 and g-mean as effectiveness measures, are presented in Figure 2. These results allow us to formulate explicit responses to our research questions. We note that results for other evaluation metrics (Recall, Precision, Specificity) are given in Appendix B.

**RQ1 – Can we obtain similar monitoring performance without assuming prior knowledge of runtime threats during threshold tuning?** As anticipated, the best strategy is ID+T, where the Optimization set closely mirrors the Evaluation set. Interestingly, the ID+T+O and ID+O

strategies consistently demonstrate statistically equivalent performance. This suggests that if one opts to utilize a large set of generic threats for threshold tuning, the inclusion of target threat data becomes useless. This is due to the fact that target threat data samples in the Threshold Optimization set are diluted among the other threats, diminishing their influence on the threshold selected.

**RQ2 – How helpful is the inclusion of generic threat data?** With OS+F1 as the effectiveness measure, the ID+O strategy outperforms ID. Conversely, with g-mean as the effectiveness measure, ID outperforms ID+O. Hence, to know precisely the benefits of incorporating other generic threats, we perform a Wilcoxon test to compare the ID strategy optimized with g-mean to ID+O optimized with OS+F1. Our results reveal that the ID strategy is superior to ID+O when evaluating g-mean scores on the Evaluation sets (p-value=3e-10) and that there is no statistical difference between the two strategies for the F1 evaluation metric (p-value=0.2). In other words, without knowledge about the expected threats a system might face, it is preferable to rely solely on in-distribution data to determine the monitoring threshold and to use g-mean for optimization.

Figure 2 also indicates that ID+T is better than ID+T+O. This suggests that supplementing the Threshold Optimization set with an arbitrary pool of threat data is not beneficial. If the target threat has been identified, it is advisable to use a combination of ID and specific threat data. Introducing data related to other random threats simply penalizes the monitor. However, it is worth noting that incorporating threats from more narrowly defined categories, closely aligned with the expected system threat, might offer improved generalization and could be explored in future research.

## 6 QUALITATIVE DISCUSSION

As anticipated, superior results were obtained for ID+T, i.e., tuning the threshold with data closely mirroring the evaluation dataset yielded the best results. However, the decreased performance observed when adding generic threats to the Optimization set is less intuitive. In this section, we propose to try to understand this behavior through an example.

To ensure that the chosen example offers meaningful insights, we select a case where the performance differences across strategies align with the conclusions presented above. For clear visualization, we require a monitor that exhibits good separability (AUROC > 0.8 on the Evaluation set), and we select the scenario that shows the maximum performance variability among strategies. Details about the chosen example can be found in Appendix C. Figure 3 shows the distributions of monitoring scores of the Threshold Optimization sets for the ID, ID+O, and ID+T strategies, as well as for the Threshold Evaluation set. The thresholds derived from both effectiveness measures are also displayed.

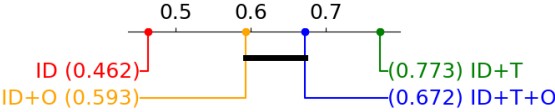

(a) F1 (effectiveness measure: OS+F1)

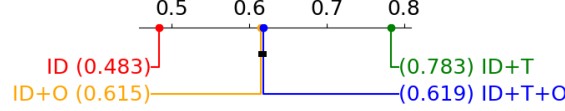

(b) g-mean (effectiveness measure: OS+F1)

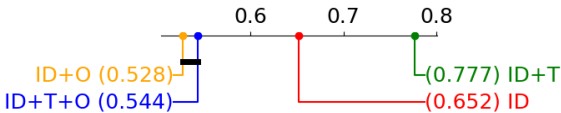

(c) F1 (effectiveness measure: g-mean)

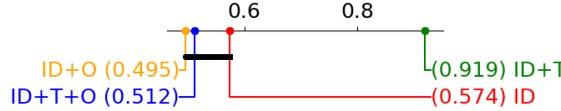

(d) g-mean (effectiveness measure: g-mean)

Figure 2: **Optimization sets comparison** – Critical distance diagram (Nemenyi test). The horizontal axis represents the average rank of the strategies. A black bar connecting two or more strategies indicates no significant difference.

Examining Figures 3b and 3d, we observe that the ID+T strategy yields score distributions most resembling those in the Evaluation set, leading to near-optimal thresholds, especially when using g-mean. In contrast, the ID strategy (Figure 3a) shows error scores (in blue) that are too close to the correct ones, resulting in smaller thresholds. However, it is worth noting that the ID strategy performs particularly well for this example, likely due to FGSM attacks generating images closely resembling the originals.

Figure 3c shows the limitations of ID+O. Interestingly, the failures differ based on the effectiveness measure used. With OS+F1, the threshold is too small because the error score distribution stretches excessively to the left. As F1 tries to minimize missed errors, i.e., maximize Recall, it pushes for a smaller threshold. Conversely, with g-mean, the threshold is excessively high because the correct score distribution stretches excessively to the right. This is due to g-mean optimization prioritizing reducing false rejections to maintain Specificity. The wide spread in ID+O scores can be attributed to the large variety of threat data, containing both correctly classified data deviating from the training distribution to imperceptible threats triggering errors.

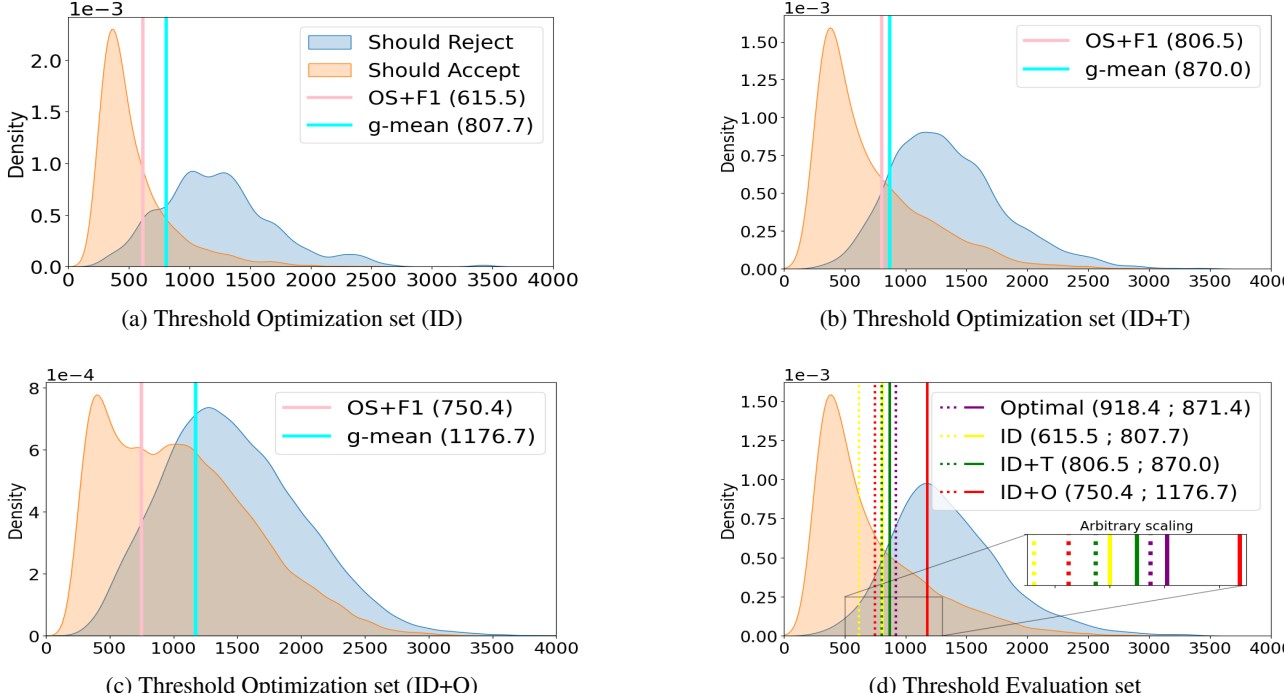

Figure 3: **Visual example to explain our findings** – Distributions of monitoring scores for the Optimization and Evaluation sets. Selected example: ID data: CIFAR10, threat: FGSM, NN: Resnet, monitor: Mahalanobis. Vertical lines represent thresholds obtained with different effectiveness measures. In (d), the dashed (resp. plain) lines represent thresholds obtained with OS+F1 (resp. g-mean). The "Optimal" thresholds maximize the effectiveness measures on the Evaluation set.

## 7 CONCLUSION

In this study, we undertook a comprehensive experimental exploration of different ways to build threshold optimization datasets for NN runtime monitoring. Our findings yielded valuable insights into the effectiveness of these approaches and their implications for real-world applications.

Our research affirmed that the ID+T approach, which leverages knowledge of the anticipated system threat to establish optimal thresholds for monitors, outperforms all other approaches. However, it is crucial to acknowledge that assuming prior knowledge of the threat is impractical for safety-critical applications, where monitors are typically designed to safeguard systems against unforeseen threats. Our findings demonstrate that we cannot expect comparable monitoring results without such prior knowledge, potentially casting doubt on the representativeness of prior evaluation results, which employed either threshold-agnostic metrics or similar data to the test set, assuming such prior knowledge.

We also investigated the inclusion of generic threat data in the threshold optimization process. Surprisingly, our experiments revealed this approach can actually compromise monitor performance. The example discussed in Section 6 suggests that incorporating data samples from unrelated threats results in overly dispersed distributions of correct and error scores, leading to suboptimal outcomes. This raises a promising avenue for future research: exploring the inte-

gration of data samples from more narrowly defined threat categories. This approach could facilitate the design of monitors tailored to specific classes of anticipated threats, such as adversarial attacks. However, its success requires the rigorous safety analysis of the system to identify relevant threats and customize optimization sets accordingly.

Furthermore, we examined the choice of effectiveness measures for selecting thresholds on the optimization set. Our findings highlight that the appropriate effectiveness measure hinges on the specific objectives of the monitor. F1 with over-sampling yields conservative monitors reducing missed errors, while using g-mean encourages higher system availability by reducing false rejections.

Our study offers a versatile experimental methodology that can be adapted to explore several other interesting questions. First, as many studies split the evaluation dataset into validation and test sets for parameter optimization, which is equivalent to employing the ID+T approach, our framework could provide deeper insights into how much monitoring techniques rely on target threat knowledge for hyperparameter-tuning. We also aim to extend these results to other tasks, such as object detection, to formulate more comprehensive and universally applicable guidelines for crafting robust neural network monitoring systems. Finally, it would be interesting to investigate whether different families of threats react differently to the proposed strategies.

## Acknowledgements

This research has benefited from the AI Interdisciplinary Institute ANITI. ANITI is funded by the French "Investing for the Future – PIA3" program under the Grant Agreement No ANR-19-P3IA-0004.

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

# Can we Defend Against the Unknown? An Empirical Study About Threshold Selection for Neural Network Monitoring
## (Supplementary Material)

**Khoi Tran Dang**[1,2]  **Kevin Delmas**[3]  **Jérémie Guiochet**[2]  **Joris Guérin**[2,4]

[1]INSA Toulouse   [2]LAAS-CNRS, Univ. Toulouse   [3]ONERA, Toulouse, France
[4]Espace-Dev, IRD, Univ. Montpellier, Montpellier, France
*tkdang@insa-toulouse.fr ; kevin.delmas@onera.fr ; jeremie.guiochet@laas.fr ; joris.guerin@ird.fr*

## A   FURTHER COMPARISONS OF EFFECTIVENESS MEASURES

In section 5, we compared two effectiveness measures for threshold tuning: g-mean (geometric mean of Recall and Specificity), and F1 (harmonic mean of Precision and Recall) complemented with over-sampling (OS+F1). Here, we extend this analysis to include F1 without over-sampling, and a typical threshold used in the literature, chosen such that the True Negative rate of the Threshold Optimization set is set to 0.95. We use the same protocol for comparison, employing the Wilcoxon signed-rank test to determine what effectiveness measure yields better performance across five evaluation metrics. We note that the performance is always evaluated on the appropriate Threshold Evaluation sets.

To confirm that the oversampling approach is beneficial, we compare using F1-score with and without oversampling as effectiveness measures. Table 3 shows the results. In general, oversampling gives better F1 and g-mean scores on the Threshold Evaluation set. We can conclude that this oversampling strategy works and reduces the bad behavior of rejecting all inputs in imbalanced scenarios (see Section 4).

Table 3: **Effectiveness measures comparison (F1 with oversampling vs. F1 without oversampling)** – Metrics were computed across the 216 experiments, followed by statistical comparison using the Wilcoxon test. The displayed numbers represent p-values, underlined orange text indicates F1 with oversampling is worse than F1 without oversampling, regular blue text indicates F1 with oversampling is better than F1 without oversampling, and italicized black text indicates no significant difference.

|             | ID      | ID+T    | ID+O    | ID+T+O  |
|-------------|---------|---------|---------|---------|
| F1          | *9e-01* | 2e-24   | 3e-12   | 3e-13   |
| G-mean      | *6e-02* | 8e-08   | 3e-23   | 6e-26   |
| Recall      | 1e-35   | 4e-06   | 7e-18   | 3e-26   |
| Precision   | 4e-35   | 1e-08   | 2e-20   | 3e-26   |
| Specificity | 1e-35   | 2e-03   | 3e-22   | 4e-26   |

In the literature, it is common to use FNR@95TNR (False Negative Rate at 95% True Negative Rate) Liu et al. [2020], Sun et al. [2021], Wang et al. [2022] as a monitoring evaluation metric. This means that the threshold is set such that 95% of correct predictions are actually accepted by the monitor. Here, we evaluate this standard literature threshold against the threshold obtained from proper optimization with g-mean as the effectiveness measure. Table 4 clearly shows that threshold optimization is better than 95% TNR for balanced metrics (F1 and g-mean). Precision and Specificity are better for 95% TNR by construction. We also note that similar results were obtained when comparing 95% TNR against OS+F1.

Table 4: **Effectiveness measures comparison (@95TNR vs. g-mean)** – Metrics were computed across the 216 experiments, followed by statistical comparison using the Wilcoxon test. The displayed numbers represent p-values, underlined orange text indicates that the metric score with threshold chosen @95TNR is worse than optimized with g-mean, regular blue text indicates that the metric score with threshold chosen @95TNR is better than optimized with g-mean, and italicized black text indicates no significant difference.

|  | ID | ID+T | ID+O | ID+T+O |
|---|---|---|---|---|
| F1 | 6e-16 | 4e-30 | 4e-37 | 4e-37 |
| G-mean | 3e-21 | 3e-37 | 3e-37 | 3e-37 |
| Recall | 3e-37 | 3e-37 | 3e-37 | 3e-37 |
| Precision | 7e-35 | 1e-26 | 1e-25 | 1e-22 |
| Specificity | 3e-37 | 3e-37 | 3e-37 | 3e-37 |

## B  FURTHER COMPARISONS OF OPTIMIZATION SET CONSTRUCTION APPROACHES

In this section, we present more performance comparisons of the four approaches for constructing the Threshold Optimization set, using other evaluation metrics (Recall, Precision, and Specificity). The results obtained with OS+F1 as the effectiveness measure are given in Figure 4 and the results obtained with g-mean as the effectiveness measure are given in Figure 5.

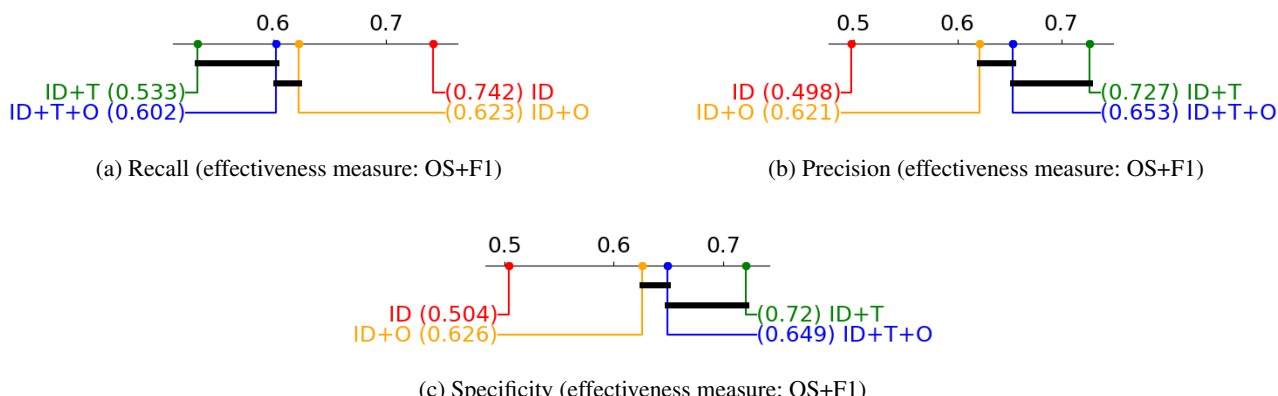

(a) Recall (effectiveness measure: OS+F1)  (b) Precision (effectiveness measure: OS+F1)

(c) Specificity (effectiveness measure: OS+F1)

Figure 4: **Threshold Optimization sets comparison, with OS+F1 as the effectiveness measure** – Critical distance diagram showing the results of the Nemenyi test. The horizontal axis represents the average rank of the approaches. A black bar connecting two or more approaches indicates no significant difference.

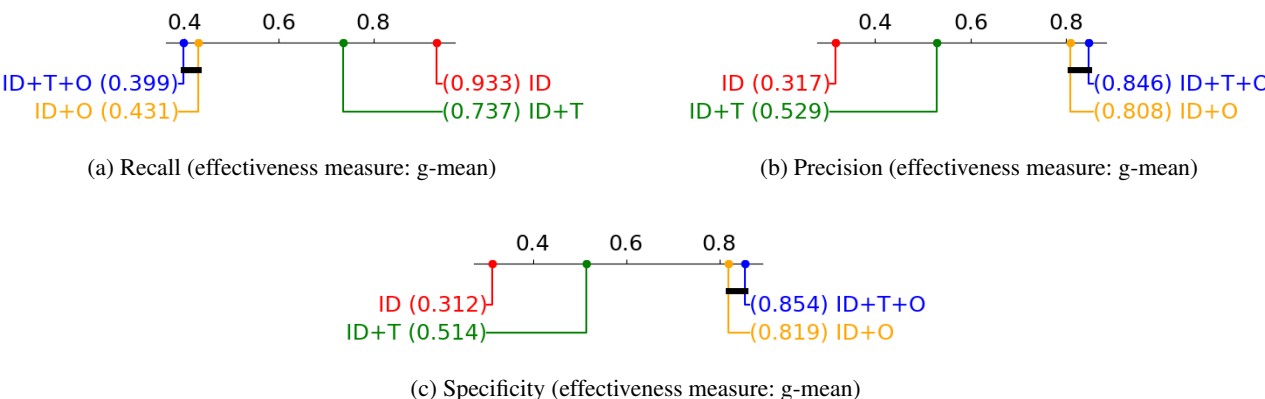

(a) Recall (effectiveness measure: g-mean)  (b) Precision (effectiveness measure: g-mean)

(c) Specificity (effectiveness measure: g-mean)

Figure 5: **Threshold Optimization sets comparison, with g-mean as the effectiveness measure** – Critical distance diagram showing the results of the Nemenyi test. The horizontal axis represents the average rank of the approaches. A black bar connecting two or more approaches indicates no significant difference.

# C  ADDITIONAL INFORMATION ON THE EXAMPLE DISCUSSED IN SECTION 6

Section 6 aims to discuss a qualitative example to illustrate and better understand the results obtained from our experimental analysis. For clear visualization, we selected a scenario that exhibits good separability (AUROC > 0.8 on the Evaluation set), and that shows the maximum performance variability among different approaches. Consequently, the selected scenario is composed of the Mahalanobis monitor, used with the Resnet NN on the CIFAR10 ID dataset, and with the FGSM attack as the threat.

Tables 5 and 6 show additional information about this example. More specifically, we present the values taken by the five evaluation metrics on the Threshold Evaluation set, as well as the AUROC score for each threshold optimization approach (ID, ID+T, ID+O), and each effectiveness measure.

Table 5: **Monitoring performances for the selected qualitative example, with OF+F1 as the effectiveness measure** – Measured metrics scores on the Threshold Evaluation set with different approaches, with OS+F1 as the effectiveness measure.

| Approach | F1 | g-mean | recall | precision | specificity | AUROC |
|----------|-------|--------|--------|-----------|-------------|-------|
| ID | 0.587 | 0.730 | 0.971 | 0.421 | 0.549 | 0.848 |
| ID+T | 0.636 | 0.787 | 0.909 | 0.490 | 0.681 | 0.848 |
| ID+O | 0.629 | 0.780 | 0.937 | 0.473 | 0.649 | 0.848 |

Table 6: **Monitoring performances for the selected qualitative example, with OF+F1 as the effectiveness measure** – Measured metrics scores on the Threshold Evaluation set with different approaches, with g-mean as the effectiveness measure.

| Approach | F1 | g-mean | recall | precision | specificity | AUROC |
|----------|-------|--------|--------|-----------|-------------|-------|
| ID | 0.636 | 0.787 | 0.909 | 0.490 | 0.681 | 0.848 |
| ID+T | 0.643 | 0.791 | 0.879 | 0.507 | 0.713 | 0.848 |
| ID+O | 0.589 | 0.719 | 0.613 | 0.568 | 0.843 | 0.848 |