# OpenReview forum: "Can we Defend Against the Unknown? An Empirical Study About Threshold Selection for Neural Network Monitoring"
_auai.org/UAI/2024/Conference — UAI 2024 poster_

### Official Review · Reviewer_XTqG · 2024-03-09

**Q2-1 Originality-Novelty:** 3
**Q2-2 Correctness-Technical Quality:** 2
**Q2-5 Clarity Of Writing:** 2

**Q1 Summary And Contributions:**

This work addresses the critical aspect of runtime monitoring in neural network-based systems, focusing on the need to reject unsafe predictions during inference. While many techniques have been developed to distinguish between safe and unsafe predictions using rejection scores, the challenge of setting an effective threshold for binary decision-making remains underexplored. Current evaluation methods, like the area under the receiver operating characteristic curve, are threshold-agnostic and do not directly address the real-world necessity of converting scores into actionable decisions. This paper highlights the limitations of assuming that runtime data distribution aligns with the training distribution, a common oversight that neglects the potential for unforeseen threats. Through comprehensive experiments on various image datasets, this study explores the monitors' ability to handle such threats beyond their training and examines the potential benefits of incorporating generic threats into threshold optimization. This approach aims to improve the robustness and practical applicability of runtime monitors in critical neural network applications.

**Q2-3 Extent To Which Claims Are Supported By Evidence:**

3: Good: the main claims are supported by convincing evidence (in the form of adequate experimental evaluation, proofs, (pseudo-)code, references, assumptions).

**Q2-4 Reproducibility:**

1: Poor: key details (e.g. proof sketches, experimental setup) are incomplete/unclear, or key resources (e.g. proofs, code, data) are unavailable.

**Q3 Main Strengths:**

1. The target problem is interesting.

**Q4 Main Weakness:**

1. This paper is not very easy to follow. For example, Fig.1 is not explained in the introduction.

2. The proposed methods are not clear. I suggest the authors provide a figure to explain the proposed methods.

**Q5 Detailed Comments To The Authors:**

See Weakness

**Q9 Complying With Reviewing Instructions:**

Yes

---

> ### Author Rebuttal · Authors · 2024-04-04
>
> Dear reviewer,
>
> First of all, thank you for the time you spent reviewing our work. Here are some responses to your concerns regarding the paper:
>
> - Regarding Fig 1, we thought that it was self contained. We added a ref to Section 3 in the intro, where the reader can find more details. Could you clarify what was not clear when reading the intro.
>
> - Regarding the addition of another figure to explain the proposed methods, we are not sure what you are talking about. Our main goal is to compare different ways to construct threshold optimization sets in order to see how much monitoring performance depends on our ability to anticipate future threats and whether adding additional generic threats could help increase robustness to unforeseen threats. The methods to construct the training sets are presented in Figure 1. As the other 4 reviewers gave us scores above 3 for clarity of writing, we would greatly appreciate if you could tell us what part of the methodology was not clear to you so that we can improve our work.
>
> - Could you explain us why you gave us such a poor rating for reproducibility? We provided an anonymized github repo with the complete code to reproduce our experiments and tried our best to describe all the hyperparameters we used. If you could provide us with specific points for improvement, we would be grateful and would try to incorporate them in a revised version of the paper. The other 4 reviewers gave us a rating of 3 or more for reproducibility.

---

### Official Review · Reviewer_nx5n · 2024-03-20

**Q2-1 Originality-Novelty:** 3
**Q2-2 Correctness-Technical Quality:** 2
**Q2-5 Clarity Of Writing:** 3

**Q10 Ethical Concerns:**

No.

**Q1 Summary And Contributions:**

The authors propose an experimental setup for evaluating runtime monitors of neural networks against various, also unforeseen, threats on computer vision datasets. They put a special focus on designing a suitable dataset for optimizing the threshold of the monitor. They do so by comparing and evaluating four different strategies of designing such a dataset. In their experiments, they find that incorporating prior knowledge about threats into the threshold optimization yields (unsurprisingly) the best monitor performance. They also find that data about generic threats might actually hinder monitor performance.

**Q2-3 Extent To Which Claims Are Supported By Evidence:**

2: Fair: the main claims are somewhat supported by evidence (but the experimental evaluation may be weak, or does not match entirely with the claims, important baselines may be missing, proofs contain important ideas but lack rigor, algorithmic details are only discussed superficially, references are imprecise, assumptions are not sufficiently motivated or explicated, etc.).

**Q2-4 Reproducibility:**

4: Excellent: key resources (e.g. proofs, code, data) are available and key details (e.g. proof sketches, experimental setup) are comprehensively described for competent researchers to confidently and easily reproduce the main results.

**Q3 Main Strengths:**

- The authors study the relevant topic of threshold fitting and evaluation.
- The introduced approaches are probably straightforward to apply in practice to evaluate real-world systems.

**Q4 Main Weakness:**

- The novelty of this work seems limited, e.g. the evaluation of the effectiveness of the different ways to construct datasets is taken from (Demska, 2006), the different ways to construct the threshold optimization datasets seems very straightforward.
- I find it hard to judge how general or robust the findings are: Are they only relevant to the CV architectures studied in the experiments, or the limited amount of attacks or types of covariate shifts? Do the results generalize to other tasks?
- No theoretical guarantees or results are provided. I acknowledge that such results are hard to get in the research area of this paper.

**Q5 Detailed Comments To The Authors:**

- I like the structure, specifically the use of the research questions that are answered using the experiments. Sometimes, the writing could be clearer, especially the experimental sections (4 and 5) I found quite hard to follow. The number of how many experiments you ran is repeated quite often.
- The related work section is quite long, could be shortened in favour of more in-depth discussion of open questions / more experiments
- Why not consider only ID data, and in another Threshold Evaluation Set only data that fits the target threat? Wouldn't this disentangle the results even more?
- In section 3, you claim that it is unlikely for a NN to encounter multiple threats at once. I think that this claim cannot be supported, especially when it comes to adversarial attacks. Why would an attacker limit his/herself to one attack when mixing multiple attacks might have a higher chance for success?
- This also holds for the threshold tuning. Wouldn't it be better to also tune the threshold against multiple attacks (if you suspect them to happen)? Can this be evaluated with your approach?
- Why did you select these specific attacks and threats in your experiments? Are they diverse enough?
- What is the null-hypothesis and significance level that you used in the Wilcoxon signed-rank test?
- I found it hard to understand what is evaluated in the different experiments (especially table 2): Are you averaging over the performance of two different NNs, 4 monitors, 3 ID, 9 threats?
- Did I understand correctly that the evaluation set is constructed in such a way that it doesn't include class imbalance?
- Why do you use two different evaluation approaches in section 5.1 vs 5.2? (Couldn't you use also Friedman + Nemenyi for comparing the effectiveness measures? Or is that just not necessary?)
- I don't think that reporting the p-values of the different experimental settings is very interpretable (e.g. table 2). I would have maybe preferred to see both average recall / F1 / etc. values, and an indication if the difference is significant. Maybe it would have also been nice to see the standard deviation of the metric values.
- Did you consider different random splits between optimization and evaluation set?
- Do you see a difference between these averaged results and the ones for specific threats? Is the problem of including generic threat data reducing performance visible over all classes of threats? I would have liked to see results splitted up in this way, e.g. in the appendix. This would potentially also allow one to judge if the presented results are skewed by the selection of threats.
- I like figure 3.
- In the appendix you claim that "In the literate, it is common to use FNR@95TNR ...", it would be nice to give some paper references as examples.

**Q9 Complying With Reviewing Instructions:**

Yes

---

> ### Author Rebuttal · Authors · 2024-04-04
>
> Dear reviewer,
>
> Thank you so much for your time and very detailed review. Here are some points to respond to your main concerns about our paper:
>
> - **Novelty**: The novelty of our approach is not in the proposed methodology but in highlighting a potential flaw in how NN monitors are evaluated. Indeed, most works evaluate their approaches with threshold agnostic metrics on a specific test set. By doing so, good results only indicate the *existence* of a good threshold for the threat being evaluated. Assuming that such results represent the monitor’s behavior in practical use cases boils down to assuming access to prior knowledge of the threat to find such threshold. Our work shows that if this assumption does not hold (which is almost always the case in practice), we cannot rely on such threshold agnostic metrics to represent accurately the monitor’s performance. To the best of our knowledge, no prior studies have pointed out this issue, which we believe is essential to develop safe NN monitors, which are usable in practice. Although it was expected that “knowledge of the target threat” would provide better results than “no knowledge of the threat”, this intuition is not being considered by most researchers from the field, who keep evaluating their approach under the invalid assumption that the threat is known in advance. We believe that our paper has the potential to raise awareness on this important issue.
> - **Generalizability of our findings**: please see response to Reviewer qJ5V.
> - **Theoretical guarantees**: To the best of our knowledge, theoretical guarantees about NN robustness and monitoring have only been achieved in simplified contexts or locally (in the neighborhood of individual samples). We do not see how to propose theoretical proofs on such complex real-world datasets.
> - **Splitting evaluation set**: Considering two separate evaluation sets (only ID and only threat) would be feasible. It would lead to even more unbalanced datasets but some metrics are robust to that problem. In this work, we studied them jointly to represent different potential failures at once (over-rejection vs over-acceptance). We believe that our approach is valid.
> - **Merging multiple threats**: It is indeed possible to encounter multiple threats simultaneously in practical scenarios. However, robustness to multiple threats can be deduced from our experiments by a simple (weighted) average of the results for each threat of interest. Regarding the possibility of tuning the threshold against multiple threats, it is precisely what has been done. The case ID+O represents threshold tuning against multiple threats, not including the target threat. The case ID+T+O represents threshold tuning against multiple threats, including the target threat. Our results suggest that, in both case, adding more threats decreases robustness, which was an unexpected result.
> - **Average results**: We did not report the average and standard deviation of the results for the same reason that we chose to use non parametric tests instead of a basic T-tests. The variability of the monitoring results obtained for different threats is too much and the results do not follow normal distributions, which is why we have to use the orderings of metrics on different scenarios, to avoid biasing the results with extreme cases (very good or very bad performance on one specific dataset). For this reason, we believe that reporting averages could lead to misleading interpretations of the results. See (Demsar, 2006) for more information.
> - **Class Imbalance**: We only modified datasets to reduce class imbalance for the optimization set in the over-sampling+F1 setting. In all the other cases, including evaluation sets, the imbalance is handled by the metrics selected (recall, specificity, Gmean)
> - **Wilcoxon test**: Significance: p-value<0.05, Null hypothesis X=Y and if rejected, we compare average ranks to know which one is best (Added in final version of the paper). The Wilcoxon test is used to compare optimization metrics because it is better suited to compare two approaches, whereas Friedman+Nemenyi is to compare 3 approaches or more. For details, see (Demsar 2006).
> - The idea of analyzing results per family of threats is one that we want to explore. We added this future work idea to the conclusion.
> - We did not consider multiple random splits because of time and compute constraints. However, we believe that the large amount of scenarios considered already accounts for most of the variability in the results.
> - We proof-read sections 4 and 5 and removed repeated occurrences of the number of experiments.
> - In Table 2 we are not averaging but conducting a statistical comparison (Wilcoxon) across the 216 scenarios experimented (2 NNs, 4 mon, 3 ID, 9 threats).
> - References added to the appendix.

---

### Official Review · Reviewer_qJ5V · 2024-03-22

**Q2-1 Originality-Novelty:** 3
**Q2-2 Correctness-Technical Quality:** 3
**Q2-5 Clarity Of Writing:** 4

**Q1 Summary And Contributions:**

In their paper, the authors discuss the application of neural networks to monitoring, i.e., observing systems with the purpose of detecting misbehavior. In particular, the authors focus on identifying thresholds with the best performance, ideally without considering assumptions. In the paper, the authors describe the underlying challenges and discuss results obtained from an experimental evaluation considering four threshold optimization techniques.

**Q2-3 Extent To Which Claims Are Supported By Evidence:**

3: Good: the main claims are supported by convincing evidence (in the form of adequate experimental evaluation, proofs, (pseudo-)code, references, assumptions).

**Q2-4 Reproducibility:**

3: Good: key resources (e.g. proofs, code, data) are available and key details (e.g. proofs, experimental setup) are sufficiently well-described for competent researchers to confidently reproduce the main results.

**Q3 Main Strengths:**

+ An interesting approach to optimization of thresholds for monitoring applications
+ The paper comprises an experimental evaluation showing the superiority of one approach
+ The content is recent and of practical interest

**Q4 Main Weakness:**

- The experiments are based on a few datasets only, which may bias the outcome
- Threats to validity are not (at least sufficiently) given but should be explicitly discussed in any paper dealing with experiments.

**Q5 Detailed Comments To The Authors:**

The paper is well-written and well-structured. The challenges, solutions, and experiments are described reasonably. The experiments revealed that there is one approach (ID+T) that leads to the best results considering different measures. The content is recent and worth publishing. It is also valuable for application in practice. There is only one drawback: The experiments are based on several datasets used in practice. However, whether these datasets allow generalizability of findings is (at least to my knowledge) not justified. Hence, it would be great to extend the evaluation considering classical monitoring applications. Besides discussing threats to validity in more detail, there are no further changes required.

**Q9 Complying With Reviewing Instructions:**

Yes

---

> ### Author Rebuttal · Authors · 2024-04-04
>
> Dear reviewer,
>
> Thank you for your time and valuable comments.
>
> Regarding the generalizability of the findings from our experiments, we tried to represent diverse scenarios while keeping the experiments tractable. While it is true that we only used three ID datasets, we tried to see the robustness of monitors to various kinds of threats, ranging from brightness changes to novelty classes and adversarial attacks. To decrease potential biases due to fixed parameters, we also compared two neural networks and 4 monitoring approaches, leading to a total of 216 distinct scenarios. We agree that having more datasets and scenarios will always lead to stronger conclusions, but we believe that our experiments already provide valid guidelines for the development of future safety monitors for NN applications. Even though they do not guarantee applicability for every future scenario, they have the benefit of raising awareness on the critical issue of threshold tuning and robustness to unforeseen threats when developing NN monitors.
>
> However, one main limitation of our experiments is that they focus on image classification, and we agree that it would be worth extending these experiments to other computer vision tasks such as object detection or semantic segmentation. Thanks for pointing that out, we added a sentence at the end of the conclusion to highlight this limitation and open new perspectives for future work.

---

### Official Review · Reviewer_n8nM · 2024-03-22

**Q2-1 Originality-Novelty:** 3
**Q2-2 Correctness-Technical Quality:** 3
**Q2-5 Clarity Of Writing:** 4

**Q1 Summary And Contributions:**

The authors investigate the problem of computing the optimal threshold for neural network monitors in the presence of out-of-model-scope (OMS) threat during test-time.  The authors investigate compare optimizing the optimal threshold on just in distribution (ID) data, ID + target threat data, ID + other generic threat data, and ID + target threat + generic threat data.

**Q2-3 Extent To Which Claims Are Supported By Evidence:**

3: Good: the main claims are supported by convincing evidence (in the form of adequate experimental evaluation, proofs, (pseudo-)code, references, assumptions).

**Q2-4 Reproducibility:**

3: Good: key resources (e.g. proofs, code, data) are available and key details (e.g. proofs, experimental setup) are sufficiently well-described for competent researchers to confidently reproduce the main results.

**Q3 Main Strengths:**

- paper is very clearly written with sufficient background
- problem is interesting and well motivated by practical application
- the empirical evaluation is useful as it tells us that in practice the threshold can be optimized best when the target data is known, demonstrating the difficulty of defending against the unknown
- experiments on multiple datasets with calculated statistical significance of findings

**Q4 Main Weakness:**

None that I can think of

**Q5 Detailed Comments To The Authors:**

Overall I thought the paper was clear and experimental results were both interesting and comprehensive.  I'm not very familiar with threshold optimization and had a question: to what extent is this observed gap between having prior knowledge vs having no knowledge due to the threshold optimization algorithm used? Is there room for improvement in terms of algorithm development or is the algorithm studied in the paper theoretically optimal?

**Q9 Complying With Reviewing Instructions:**

Yes

---

> ### Author Rebuttal · Authors · 2024-04-04
>
> Dear reviewer
>
> Thank you for your thoughtful review. We're pleased you found our paper clear and the experiments comprehensive.
>
> To answer your question, the concept of threshold optimality can be defined in different ways. The thresholds that we use in our experiments are optimal with respect to the Threshold Optimization set (what we have for training), but they are often not optimal on the Evaluation set (which is unknown when fitting the threshold). Optimality on the Optimization set is guaranteed because we conduct an exhaustive search on a discrete dataset.
>
> Other threshold optimization algorithms would either try to approximate this optimal threshold with less iteration to increase efficiency, or would need to assume knowledge of the underlying distribution which generated the Optimization set, thus transforming threshold tuning into a continuous problem.
>
> In this work, we do not assume knowledge of the underlying optimization set distributions, and we do assume knowledge of the data encountered at test time. These choices aim to mirror what happens in practical scenarios, where we do not know what threat a system will face during inference. Under these assumptions of limited information for training, an exhaustive search on the optimization set is the best that can be achieved.

---

### Official Review · Reviewer_C8yt · 2024-03-24

**Q2-1 Originality-Novelty:** 1
**Q2-2 Correctness-Technical Quality:** 3
**Q2-5 Clarity Of Writing:** 3

**Q1 Summary And Contributions:**

This paper studies thresholding optimization in runtime monitoring. Authors conduct experiments on image datasets to investigate the effectiveness of monitors in handling unforeseen threats that are not available during threshold adjustments, and whether integrating generic threats into the threshold optimization scheme can enhance the robustness of monitors. Writing is very clear.

**Q2-3 Extent To Which Claims Are Supported By Evidence:**

3: Good: the main claims are supported by convincing evidence (in the form of adequate experimental evaluation, proofs, (pseudo-)code, references, assumptions).

**Q2-4 Reproducibility:**

3: Good: key resources (e.g. proofs, code, data) are available and key details (e.g. proofs, experimental setup) are sufficiently well-described for competent researchers to confidently reproduce the main results.

**Q3 Main Strengths:**

Presentation is good, with sufficient and accurate descriptions of background and the used metrics.

**Q4 Main Weakness:**

The topic and the approaches do not provide exciting insights.

**Q5 Detailed Comments To The Authors:**

The paper writing is clear and I only have a concern on the contribution of the paper. To me, an empirical evaluation paper can be accepted if it conducts very through experiments and provides many new insights (potentially with theoretic analysis). However, the current results and approaches, e.g., investigating threshold using different datasets (ID, ID+T, ID+T+O, ID+O) are somewhat straightforward and the two RQs are expectable. And there is no further consequence of these findings, e.g., deeper analysis or application to real cases.

Overall, I do not think the contribution meets the bar of a full paper. A suitable workshop may be a good fit.

**Q9 Complying With Reviewing Instructions:**

Yes

---

> ### Author Rebuttal · Authors · 2024-04-04
>
> Dear reviewer,
>
> First of all, thank you for the time you spent reviewing our work.
>
> Here are several points to respond to your concerns about the paper:
>
> 1. We do not agree that the answers to our research questions are expectable. When we started conducting these experiments, we were convinced that training a monitor on a pool of potential threats would help to enhance its robustness to new threats, as it is usually the case for most learning systems. However, our experiments showed the opposite behavior. The fact that using merely ID data to fit the monitoring threshold, leads to monitors that are more robust to unforeseen threats was unexpected and has very practical implications for real-world use cases.
> 2. In practice, in the NN monitoring literature, people train their monitors on ID data and test them on specific threats, using threshold agnostic metrics. By ignoring the problem of threshold tuning, the results reported in the literature are highly over-optimistic. By presenting the results of our experiments to the community, we hope to shed light on this issue and motivate a change in NN monitor evaluation methodology, by including some criteria regarding the usability for unforeseen threats.
> 3. While we acknowledge the value of theoretical analysis, we argue that the complexity and unpredictability of real-world scenarios often necessitate empirical evaluations. Our rigorous experimental methodology and statistical analysis furnish the community with practical insights that simplified theoretical models may not accurately capture.
>
> Additionally, it's worth mentioning that the question of our work's novelty appears to be a point of contention solely in your review, as other reviewers have not expressed similar concerns. We believe this diversity of perspectives underscores the subjective nature of evaluating novelty, which depends highly on the reader’s interest. However, we hope that the explanations provided above clarify the unique insights and value our work brings to the field.

---

### Meta-Review · Area_Chair_BBPs · 2024-04-17

This paper performs an empirical study of how to choose a specific threshold for rejection / abstention, as most prior work has been evaluated with threshold agnostic metrics, like AUC.  The reviewers had mixed opinions of the work (3 x borderline reject, 2 x accept), with the primary criticisms being about sufficient novelty and if the scope of the experiments is sufficient.  After reading the discussions and the paper itself, I believe the paper speaks to a very important practical question (e.g. how to choose a threshold) with sufficient experimental rigor.  Regarding novelty, I find the paper's experimental evaluation to be a sufficient contribution, even though no new method is proposed.  Regarding scope, perhaps the paper should take more care to qualify that the findings only apply to computer vision, but with that caveat, I find the experiments sufficient and interesting.  Furthermore, I find the paper's writing and presentation to be admirably clear.

For revisions, I ask that the authors take care to make explicit the scope / limitations of the experimental conclusions.  I also urge the authors to re-consider the paper's title, as it is overly broad and non-specific.  Personally, I think something like "Can We Defend Against the Unknown?: an empirical study of threshold selection" would better attract the attention of practitioners who have the difficult job of selecting a single threshold.